# Participatory Guarantee Systems: When People Want to Take Part

**Mamen Cuéllar-Padilla [1], Isabel Haro-Pérez [1,*] and Mirene Begiristain-Zubillaga [2]**

[1]   Agroecology, Food Sovereignty and Commons Research Group, Sociology Unit, C5 Building, Campus Universitario Rabanales, Cordoba University, 14080 Cordoba, Spain; mcuellar@uco.es

[2]   Department of Finance Economy II, Faculty of Economics and Business, University of the Basque Country (UPV-EHU), Plaza de Oñati 1, 20018 Donostia-San Sebastián, Basque Country, Spain; mirene.begiristain@ehu.eus

*   Correspondence: isabel.haro@uco.es

**Abstract:** Participatory guarantee systems (PGSs) have emerged as a response to exclusion and intermediation processes involving third-party certification, which is currently the only guarantee system recognised by the European Union for organic food. Despite their unofficial recognition, PGSs are developing and generating shared frameworks of action. In this research, through three certification bodies (two public and one private) and eight PGSs in Spain, we investigate the similarities and differences between the procedures and tasks that both systems develop in order to generate trust in the decision-making structures involved and the responsibilities on which they are based. While the overall organisation of the systems is very similar, there are profound differences in their decision-making: their procedures and who participates in them. The differences we highlight lead us to argue that PGSs effectively solve the exclusion problems that third-party certification generates. Specifically, they offer lower costs and more accessible bureaucracy. They also generate and strengthen, through trust-building, the links and processes of local self-management and empowerment. However, developing PGSs demands much time and dedication, and their official regulation is complex, so it is difficult to predict that they will be widely adopted.

**Keywords:** third-party certification; procedures; participatory governance systems; food system democratisation; alternative food networks

## 1. Introduction

In recent years, participatory guarantee systems (PGSs) have been the subject of a growing, although still emerging, scientific output. Various articles have been published analysing them from the perspective of their operating mechanisms [1–4], their strengths and benefits [4–9], and their weaknesses and challenges [1,2,10–13].

Among the operating mechanisms identified are key factors such as peer review visits and collective decision-making by stakeholders for the endorsement of membership of new entrants to the system. Numerous advantages are recognised relative to third-party certification, particularly their lower costs, their simpler administration procedures—which make the mechanisms better adapted to small-scale production—and the aspects of their procedures that generate articulation, empower small producers and strengthen the local social fabric. Two of the weaknesses identified are the complexity of their collective procedures and their diversity and heterogeneity. Another is the difficulty of making members participate in the manner required, as is also the case in other collective initiatives related to alternative agri-food systems [14]. The literature also often highlights problems such as the reliability of these mechanisms and their lack of profile in many countries.

This literature also reflects the criticisms directed at third-party certification [4,9,15–18]. One of the widely identified issues is the costs and paperwork associated with this mechanism, which make it difficult for small producers to join. Others are its underlying exam logic (pass or fail), its inefficiency in identifying fraud, the lack of differential treatment between agribusiness and small producers and the intermediation logic reflected in this procedure. The literature on PGSs is also closely linked to that on alternative food networks [19,20] and the political dimension of agroecology [21,22]. The literature also recognises in particular that the movements generated through the PGSs offer still wider benefits. They strengthen both networks and agroecological political coherence and comprise both producers and consumers willing to play a role in revolutionising food systems, aiming to create a future of organic food which is more ambitious than the current organic certification system [23].

In most cases, PGSs are presented as necessary and interesting alternatives to third-party certification, proposed as a response to its limits in contexts such as small farm holdings, marginalised territories, situations of food insecurity, and difficulty of access to markets [1,9,11]. Since PGSs pre-date third-party certification, however, it is clear they did not arise as an alternative to the latter. While the official regulations which drove third-party certification towards organic production emerged at the end of the 1980s and the early 1990s, the first systematised and internationally recognised PGSs started functioning in the 1970s [24].

One central differentiating element between the two systems is the procedures through which a guarantee is established. These cover the mechanisms, activities and decision-making processes that each system advocates in order to generate confidence. The PGS's participatory is said both to avoid most of the limits and weaknesses of third-party certification and to offer an alternative to farmers at risk of exclusion from the third-party certification system.

But what are the procedural differences between third-party certification systems and PGSs? And what does the participation approach imply for farmers and other actors involved in PGSs? Is it just an issue of size or socioeconomic profile, simply farmers preferring one system over another?

The aim of this study is to make a rigorous comparison between the procedures developed by third-party certification systems, based on Regulation 834/2007 and the Standard UNE-EN-ISO/IEC 17065, and the mechanisms developed by PGSs based on the experiences of the relevant bodies and people in Spain. We aim to evaluate how both systems build trust (the type and design of procedures) and to discuss a hypothesis that follows from two research questions. The questions are:

1.　What are the similarities and differences between the procedures of third-party certification systems and those of PGSs?
2.　What requirements should a group of producers be aware of before becoming involved in a PGSs, bearing in mind its participatory approach?

The hypothesis is twofold, as follows:

(i)　The benefits of the alternative to third-party certification offered by PGSs range wider than simply the problems of exclusion created by third-party certification, including particularly to small and medium-sized entities and in contexts of difficult access to markets).
(ii)　A PGS may offer an alternative proposal, but it calls for specific requirements and procedures from its potential members, so it might not be a valid option for every producer looking for a guarantee system.

## 2. Context and Methods

PGSs have been developing in Spain since 2004 when they emerged in the south following a policy initiative by the Andalusian government [25]. In 2015, the first meet-

ing of PGSs in Spain was organised, in which 8 PGSs participated. Since then, three more meetings have taken place, as shown in Table 1.

**Table 1.** PGSs meetings in Spain organised since 2015.

| Year | Place | PGSs Participating |
|---|---|---|
| 2015 | Valencia | A Gavela, Pontevedra//Basherri Sarea, Guipúzcoa<br>Ecollaures, Valencia//EcoRed, Aragón<br>FACPE Ecovalle, Andalucía//Red Agroecológica de Cádiz<br>SPG Xarxa Llauradora +Bo//Vecinos Campesinos, Murcia |
| 2016 | Vigo | A Gavela, Pontevedra//ASAP Castilla y León<br>Ecollaures, Valencia//EcoRed, Aragón<br>Ecovalle, Granada//Red Agroecológica de Cádiz<br>SAES, Madrid//Vecinos Campesinos, Murcia |
| 2017 | Madrid | A Gavela, Pontevedra//ASAP Castilla y León<br>Ecollaures, Valencia//Ecomercado de Córdoba<br>Ecored, Aragón//EhKolektiboa, Euskalerria<br>FACPE El Encinar, Andalucía//Mosaics de Vida, Castellón<br>Red Agroecológica de Cádiz//SAES Madrid<br>Vecinos Campesinos, Murcia |
| 2018 | Granada | Asociación Como de Graná: eSPiGa//Ecollaures, Valencia<br>EhKolektiboa, Euskalerria//El Encinar, FACPE, Granada<br>La Ortiga, FACPE, Sevilla//Mercado da Terra, Lugo<br>Mosaics de Vida, Castellón//Nature et Progrès<br>PGSs team IFOAM//Red Agroecológica de Cádiz<br>SAES Madrid//SPG Alpujarra y Costa granadina<br>Vecinos Campesinos, Murcia |

Source: Compiled by the authors.

The official certification system in Spain does not recognise these initiatives: its guarantees come from private or public bodies that, depending on the region, develop the certification. For example, in Andalusia, a private system was established in 2002. There are currently 11 private certification bodies operating in the whole area, while elsewhere, a public organic certification committee is responsible for building consumer confidence in the organic sector. Two regions are exceptions, where they have established a hybrid system in which both private companies and a public committee can certify.

The research for the present paper included the compilation and systematisation of the procedures implemented in Spain by three third-party certification bodies (two public and one private) and 8 PGSs, all of which have been operating for more than five years and have participated in at least one of the meetings organised since 2015 (shown in Table 2).

**Table 2.** Cases included in the research.

| Type of Guarantee System | Code * | Nature or Region | Year Activity Began |
|---|---|---|---|
| Third-party | TP1 | Public | 2008 |
| | TP2 | Public | 2007 |
| | TP3 | Private. Andalusia and Castile-La Mancha | 2002 |
| PGSs | PGS1 | Region of Valencia | 2012 |
| | PGS2 | Community of Madrid | 2014 |
| | PGS3 | Granada and Almeria | 2010 |
| | PGS4 | Seville and Cadiz | 2011 |
| | PGS5 | Sanlúcar de Barrameda, Cadiz | 2012 |
| | PGS6 | Murcia and Alicante | 2011 |
| | PGS7 | Basque Country | 2014 |
| | PGS8 | Galicia | 2011 |

Source: Compiled by the authors. * A code is assigned to every case studied in order to guarantee confidentiality.

For all case studies, we carried out an extensive bibliographic review of existing documentation on the guarantee systems, such as internal regulations, operating manuals and protocols on procedures, documents, and reports. In addition, the websites and blogs of the case studies were reviewed. Key informants in every case study were contacted for access to this documentation. Ongoing communication was established to clarify concepts, documents, formats and structures, and details about procedures not described in the documents.

The case study research methodology is qualitative; it involves the investigation of a real-life situation which, because of its critical analysis, can provide enough insight to generate a theoretical sample from which conclusions can be drawn [26]. Its validity does not lie in a probabilistic sample for the generalisation of results but in the development of a theory that can be transferred to other cases. For this reason, some authors prefer to talk about transferability rather than generalisation in qualitative research [27,28]. Case studies allow us to describe the real context, evaluate results, explore situations and even explain some causal relationships that are too complex for research strategies carried out using surveys or experiments. In addition, triangulation—using multiple sources of evidence—was achieved by a review of the relevant scientific literature and by analysis of the actions taken in the meetings mentioned above and during the research process, as shown in Supplementary Materials Table S1.

Once an initial review of the different procedures had been completed, we compiled a table of variables using OpenOffice Calc software to facilitate the organisation of all the data, and we encoded the information provided by the case studies, following these variables and using the Atlas.ti 9 software.

In the codification process, when inputting the information for the different third-party and PGSs procedures, we identified weaknesses and improvements that needed addressing in the original codex table. A group discussion of the exercise, which included members from some of the PGSs studied, resulted in a final table and codex model. Here, the procedures and tasks of the PGSs and third-party certification bodies were entered and organised into three types of procedures and 17 types of tasks (shown in Table 3).

**Table 3.** Procedures and tasks systematised to organise the information.

| Procedure | Tasks | Explanation |
|---|---|---|
| Entry Procedure | Initial Contact and Entry Procedure | First steps: a request for admission by the producer; the request and receipt of the documentation required and payment of the fee. |
| | Commitment declaration | Declaration of commitment signed by the new producer stating that they are familiar with the rules of the system and how to manage the farm/plots/initiative in line with these rules. |
| | Self-evaluation | Form to be filled in by the new producer, about the project's management and design. |
| | Audit/Initial visit | The first visit to the new farm/initiative. A visit checklist is completed during the visit, and a visit report is prepared soon afterwards. Soil, water or plant samples are taken when necessary. |
| | Transition period for integration into the PGS | The time that must elapse before an entity can use the label varies with the system, from direct entry to 3 years, depending on various factors. |
| | Decision and communication of the entry of the applicant | When and how the decision is taken on the final acceptance of a new producer. |
| Follow-up and Trust-Building Procedure | Audit/Follow-up visit | Audits, or follow-up visits to farms every one or two years. A visit checklist is completed during the visit and a visit/inspection report is prepared soon afterwards. |
| | Analytics | Soil, water or plant samples are taken when suspicion or doubt is raised and when high or medium risk levels are detected. They are also taken randomly. |
| | Evaluation of visits | Assessment and evaluation of the visit/inspection report. |
| | Membership renewal decision | Decision taken on the renewal of the authorisation of the use of the label/full membership of the initiative. Improvement recommendations are made in the case of PGSs. When the decision is unfavourable, sanctions are applied. |
| | Confirmation of compliance | Document valid for one year in which the farmer guarantees compliance with the norms established. |
| | Field notebook | Document that farmers must fill in detailing all the practices and activities carried out on the farm. |
| Organisation | General assemblies | Meeting to be attended by all members where the main procedures of the system are discussed and decisions are taken. This is usually annually, twice a year or by request from the different working groups |
| | Working groups | Spaces for participation and decision-making on sectoral or thematic issues. Their composition and tasks are defined by the general assembly, and outputs are passed on to, and reviewed by, the general assembly. |
| | Visiting groups | Task associated with participating in the visits carried out on the farms every year: organising and completing the visit, completing the visit checklist and writing the visit report. |
| | Fees | Annual amount of money producers must pay to take part in the system. |

Source: Compiled by the authors.

Due to a lack of official regulation of PGSs, there is no universal umbrella framework covering all PGSs, so they do not all share the same procedures. A methodological decision was therefore taken to assume that a procedure forms part of the framework of a PGS if it is shared by at least 5 out of the 8 PGSs studied.

In contrast, third-party certification bodies do have an umbrella framework that guides the development of their activities. The documents used are the Standard UNE-EN-ISO/IEC 17065, which establishes how third-party certification bodies must work, and the information provided by the certification entities detailed in Supplementary material 1. In this case, despite our attempts to access this type of documentation from other private certification bodies, only one existing enterprise operating in Andalusia agreed to provide such documents. The other informants said that it could not be provided because it was confidential information about its internal affairs, despite the research group offering to sign a confidentiality contract. However, due to the common official regulation framework, and after the similarities between the three certification bodies results obtained, we can affirm that the data systematized are consistent.

In Figure 1, a general schema of the methodological process explained above is presented.

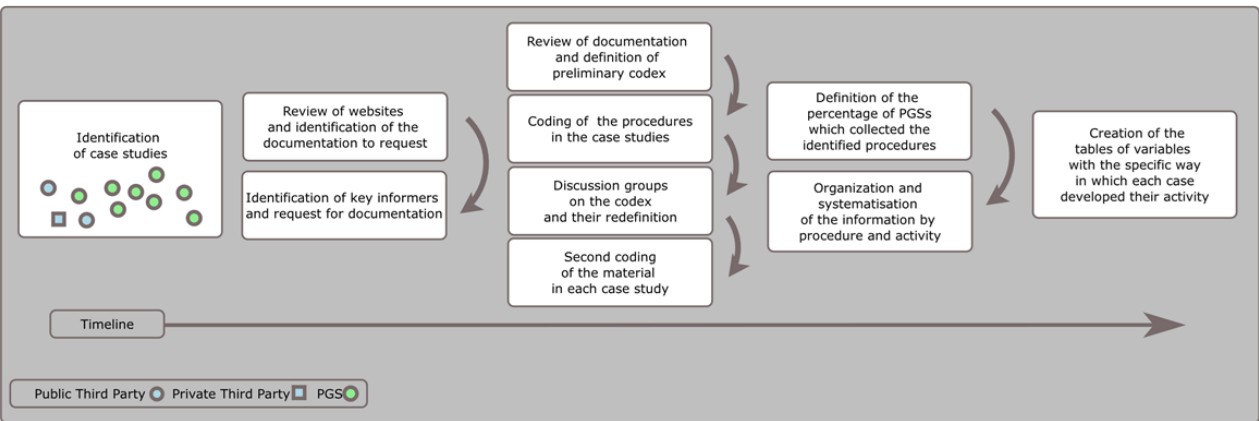

**Figure 1.** Diagram of the methodological procedure followed.

The information organised in the above table guided us in our discussions on the main similarities and differences between the confidence-building schema proposed by PGSs and that established by the European regulation, and on the requirements, specific developments and procedures required in both systems.

## 3. Results

By identifying the case studies and the variables they incorporate, our results begin to highlight mechanisms shared by all the PGSs analysed (even though each one has been developed by different people in a different context and without an official framework), and other mechanisms implemented only by some of the PGS.

One issue highlighted by the study is the difficulty in accessing documentation from private certification bodies, drawing attention to the non-transparency of these entities.

The other main results obtained in the research are presented in the following order: a. those related to the design of the general procedure of both PGSs and third-party certification; b. the actors involved in the procedures and tasks of both systems; c. the PGSs' governance system and the third-party certification system established to organise and execute their respective procedures and tasks.

*3.1. The General Design of the Procedures in Both Systems*

The two guarantee systems both have two main procedures. The first is the building of confidence around a new producer who wants to be part of the guarantee system for the first time. The second is a follow-up and trust-building, i.e., how the system bolsters the guarantee for the producers over time.

Every procedure contains several tasks, which may or may not be included in the guarantee systems studied. Table 4 shows whether the official certification (known as third-party certification) includes and carries out such tasks and the number of the PGSs analysed that undertake each specific task.

**Table 4.** Tasks carried out by the different guarantee systems studied.

| Procedure | Task | Number PGS | Third-Party (Yes or No) |
|---|---|---|---|
| Entry Procedure | Initial Contact and Entry Procedure | 8 | Yes |
| | Commitment declaration | 7 | Yes |
| | Self-evaluation | 2 | Yes |
| | Audit/Initial visit | 8 | Yes |
| | Transition period/PGS integration | 4 | Yes |
| | Decision and communication on the entry of the applicant | 8 | Yes |
| Follow-up and Trust-Building Procedure | Audits/Follow up visits | 8 | Yes |
| | Analytics | 5 | Yes |
| | Visits evaluation | 8 | Yes |
| | Renewal decision | 8 | Yes |
| | Confirmation of compliance | 5 | Yes |
| | Field notebook | 0 | Yes |

Source: Compiled by the authors.

In the entry procedure for both guarantee systems, a mechanism is established that allows new producers to make initial contact and request entry. Both systems then follow up with an initial visit and ask the new producer to sign a letter of commitment acknowledging the principles and standards of the guarantee system to be upheld.

An important distinction of third-party certification is that it always establishes a self-evaluation document for the person/company to be certified in addition to setting the transition/conversion period, clearly defined by the legislation, which is a period during which the producer is audited and monitored but cannot yet use the accredited quality label. This conversion period may be made shorter than that dictated by the regulations and is fixed by the certification body itself according to its evaluation of the supporting arguments submitted. In contrast, in PGSs, the self-evaluation document is not widely used and nor is the period of transition between the first visit to an initiative and its effective entry. However, this varies between cases. Just over half establish a transition period, while the rest assume that if the initial visit is positive, the producer is now a full member of the PGS. In some cases, this is because, in order to access the initiative, it is necessary to have the official seal (the farmer must already be certified by a third-party entity). In those PGSs that do establish a transition period, it ranges from 6 months to 1 year.

With regard to the endorsement follow-up, once the producer or farm is registered and authorised in the initiative, both types of scheme (PGSs and third-party certification) establish similar activities in all cases studied: visits to the farms, evaluation of these visits and decision-making about whether or not to renew the guarantee. Both

third-party certification and PGSs set regular visits, which are evaluated using a visit checklist, which is considered in detail before a final decision is made on whether to renew the registration. Third-party certification establishes analytics as a regular monitoring mechanism, and this is also the case for most, but not all, PGSs. It is worth noting that PGSs do not require a field record book to be kept (which is called a field notebook in third-party procedures).

Another important issue concerns the documents used by each system. The third-party system requires more documentation of different types and the use of a different language than that of PGSs. The third-party certification uses the language of public administration and administrative processes, written by technical staff based on the legal system it operates in. The PGSs' documents are developed by people on the ground, the producers and, in some cases, the consumers. This results in them being more accessible and understandable to non-technical people.

### 3.2. Actors Involved in the Procedures and Tasks of Both Systems

Although the frameworks presented above for the procedures of both systems are not dissimilar, it is useful to collect and analyze the results of how the process of each activity in each system is developed. This information is shown in the following figures (Figures 2 and 3) and in the additional information in Supplementary Materials Tables S2 and S3.

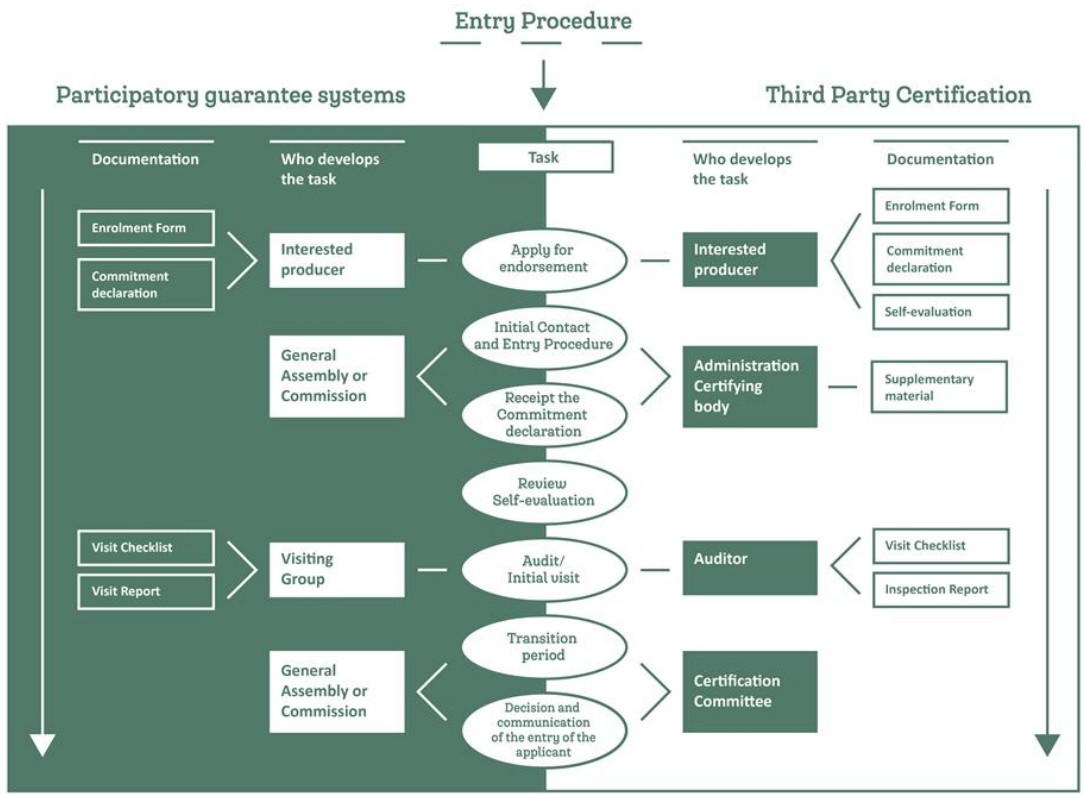

**Figure 2.** Development of the entry procedure in third-party certification and PGSs.

It is important to highlight, first, the differences in the dialogue conducted with the person seeking the guarantee. Third-party systems establish technical contact through their offices and websites, while PGSs function through direct contact between the applicant and one or more members of the initiative, in most cases other producers, or through a commission or the general assembly. Some PGSs have established a Sponsor, who is always a producer, to help the new producer throughout the entry procedure.

The new producer is, therefore, offered support from another producer in the first months of working with the PGSs.

The second key differentiating factor is how they deal with the initial visit. In PGSs, the visit is a peer-review visit carried out by various members of the initiative, with other producers always present and, on occasions, consumers and/or people with a technical background. A minimum of two producers participate in each visit, with at least one from the same sector as the applicant. The only exception to this is when the applicant producer belongs to a sector that is new to the PGS. In all cases studied, the people who participate in the visits are from the same geographical area as the applicant. This is quite different from third-party certification, where the visit is carried out by a single person with a technical background, and this same person usually covers several geographical areas.

The third important difference between the two systems is in how the final decision is made on whether or not to accept the new applicant. In PGSs, this decision is taken collectively by members of the initiative. This is a group of individuals, on occasion the general assembly itself, which can be attended by producers who did not participate in the visit and by other members of the initiative such as consumers or technical support personnel. In all cases studied, these are people from the same geographical and social context as the person being evaluated. In the case of third-party certification, this is decided by the appropriate body of the organisation, made up of one or several people who did not take part in the visit, all with a technical background.

Many of the differences identified in the entry procedure also apply to the endorsement follow-up and ongoing trust-building procedure, as shown in Figure 3 (additional information in Supplementary Materials Table S3.

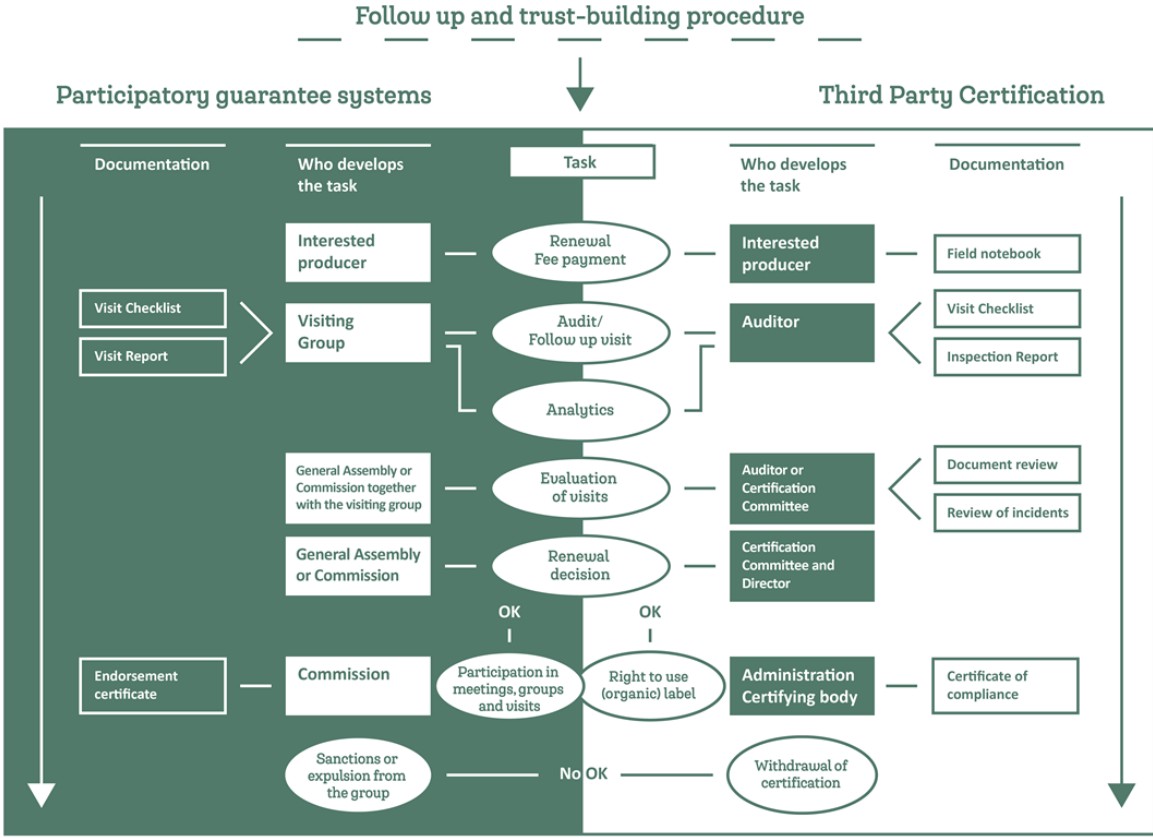

**Figure 3.** Development of the follow-up and trust-building procedure in third-party certification and PGSs.

While in both systems the basic guarantee procedure is built around three events (the visit, the evaluation of the visit, and the decision on whether or not the producer/farm visited receives the endorsement), the two systems carry out these tasks in very different ways.

In the case of ongoing visits, the differences are similar to those noted in initial visits. In PGSs, they are undertaken by other producers and, where appropriate, consumers or technical staff from the initiative. Members of the initiative are committed to participating in a minimum number of visits per year, or being a member of the visiting group for a whole year, on a rotation basis. In other words, everyone who is visited will, in turn, carry out visits accompanied by other people since visits are collective in all cases. In contrast, third-party certification systems are restricted by law from acting in an advisory role, and the technical person who carries out the visit does so purely as an auditor.

The second screening used by both systems is the evaluation of the visit report. In PGSs, this is also a peer review and includes other producers and consumers or people with a technical background who were not present during the visit. On occasions, the people who did visit the farm, and the producer being assessed, offer relevant explanations and clarifications to those who did not attend. In third-party certification, this second screening may be based on the work of a certification commission, whose membership is not made clear in the documentation that is reviewed and in some cases by a different technical person in the same company. In other words, the screening is completed by one person or several, all of whom have a technical background and evaluate visits from several geographical areas.

A key differentiating factor is that PGSs evaluate not just the latest visit but also visits from previous years. In this way, the evolution of the entity's management is assessed, and any improvement recommendations identified on previous visits can be followed up. This shows that the visits are occasions for the exchange of information, advice and knowledge between producers, leading to farm-management improvement recommendations that will be actioned in subsequent years. Whether or not these recommendations are heeded by the farm under evaluation is also a criterion for assessment and will influence the final decision. This is not the case in third-party certification; the person visited is evaluated based on the annual report, with a logic of inspection, and the result is simply a pass or fail.

Another difference in some PGSs is related to the structure and the people who make the final decision. In these cases, once the visit report has been evaluated by the appropriate commission, the final decision is made by a large collective body, such as the general assembly, in which everyone in the initiative participates. In third-party certification, this third screening does not exist.

### 3.3. The Governance Structure of the Guarantee Systems

Another difference between PGSs and third-party certification is seen in the organisation and governance of the procedures developed to build confidence and guarantees (Table 5). All PGSs establish operating procedures that require the active participation of members and establish governance systems both horizontal and participative (annual assemblies and working groups whose composition usually is on a rotation basis). None of this is considered in the procedure for third-party certification, with the people certified playing a passive role.

**Table 5.** Participatory tasks developed in the organisation structure by the different guarantee systems studied.

| Procedure | Task | PGS | Third-Party |
|---|---|---|---|
| Organisation structure | General assemblies | 8 | No |
| | Working groups | 8 | No |
| | Participation in visits | 8 | No |
| | Fees | 8 | Yes |

Source: Compiled by the authors.

Given their internal organisation, PGSs call for more active participation by their members in the different spaces of governance (general assembly, working groups). Similarly, in all the PGSs studied, participation in visits and some meetings is obligatory for their members. The main organisational structures are collected and described in Table 6.

**Table 6.** Main organisational structures in PGSs.

| Case | Structures | Description |
|---|---|---|
| PGS1 | General assembly | Four per year. Participation in at least two assemblies is compulsory. |
| | Working groups | Permanent groups: Entry commission (develop the tasks associated with the entry of new members) Quality commission (evaluate the self-evaluation forms, review the visit reports, receive any comments from the visiting groups, and resolve any doubts or conflicts during the visits). Updating commission (organise and update all the information generated in the assemblies, groups, visit reports, etc.) Food processors commission (specific working group for those members who are food processors) Special groups: Commitment to local collective organisations/Maintenance of website/"Care" Commission/PGSs research project/Organisation of annual team away-times/Translation of documents |
| | Visiting groups | Organised by the quality commission. Composed of at least one producer from the same sector as the farmer being visited. |
| PGS2 | General assembly | Four times per year. |
| | Working groups | One permanent group that organises the tasks and activities to be implemented. |
| | Visiting groups | Composed of three people (one farmer from the same sector, another member and a third person from the group or external to the PGS). |
| PGS3 | General assembly | At least one per year. Compulsory attendance of all members. |
| | Working groups | PGSs Group. Organise everything related to the development of the PGS. The animateur is a hired technical person. Composed of all the farmer members, two consumers, the hired technical person, and the person responsible for collaborative retail sales |
| | Visiting groups | Six visits per year. Farmers must participate in a minimum of three visits per year. |
| PGS4 | General assembly | At least one per year. |
| | Working groups | Permanent: Quality commission (evaluation of new members, organisation of visits and analytics, visit report evaluation, recommendations and proposal of sanctions, and follow-up). |
| | Visiting groups | Includes one farmer, one consumer and, if possible, a technician. |
| PGS5 | General assembly | Once a year. Compulsory attendance for all members. |

| | | |
|---|---|---|
| | Working groups | Permanent: Quality commission (evaluation of new members, organisation of visits and analytics, visit report evaluation, recommendations and proposal of sanctions, and follow-up). |
| | Visiting groups | Includes one farmer, one consumer and, if possible, a technician. This is organised by the quality commission. Every member must participate in at least one visit per year. |
| PGS6 | General assembly | 3 per year. Attendance of least 1 per year is compulsory. |
| | Working groups | None at present |
| | Visiting groups | A farmer nominated as the Sponsor will be part of the visiting group for all first visits. Follow-up visits: composed of three people (a farmer, a consumer and a distributor). |
| PGS7 | General assembly | In process of definition |
| | Working groups | Permanent: Sectoral groups Social issues |
| | Visiting groups | Every member must participate in at least one visit per year. The group is composed of a farmer from the sectoral group, a consumer and an animateur. |
| PGS8 | General assembly | Attendance is compulsory for all members. The periodicity is not defined. It is organised when requested by individual members or by a working group. |
| | Working groups | Permanent: Technical team (a farmer, a consumer and a technician), in rotation every year with participation being voluntary. The team is in charge of managing the visits and reports and communicating them to the assembly. Dissemination team (two producers) |
| | Visiting groups | Composed of the technical team |

Source: Compiled by the authors.

These governance instruments are not applied in third-party certification, meaning that the people certified do not participate in the process of governance related to the guarantee system.

Lastly, fees feature in both mechanisms, and an annual fee must be paid in order to receive the guarantee from the initiative. Nevertheless, there are significant differences here, too (Table 7).

**Table 7.** Annual fees established by each guarantee system (data for 2020).

| Guarantee System Case | Fees |
|---|---|
| TP1 | Two parts to the fee: Audits and certification costs: 90 € for farmers Promotion and dissemination activities: 40 € × a variable depending on the size of the farm |
| TP2 | A fixed cost (148.76 € in 2020); increases according to the type of product and the size of the farm. The cost increases according to the level of risk of non-compliance. Some bureaucratic procedures increase the cost: translation of documents, amendments to legal status, requests for reduction of the transition period, non-compliance, analytics etc. |
| TP3 | The fee is set according to the applicant. A transparent costs table is not made available, with the fee being calculated separately for each applicant. The cost of certification varies with the type of production, the characteristics of the activity, the land and the facilities. There are two types of fees: Enrolment Fee and Renewal Fee. Costs are added to these fees based on the following items: sending documentation by ordinary mail or courier, additional unscheduled visits, sampling and analytics, issuance of Certificates of Conformity, processing of Authorisation Requests and processing of modifications. |

| PGS1 | Individual projects: 20 €<br>Collective projects (mínimum two people): between 20 € and 50 €<br>Entities and collectives: between 50 € and 100 € |
|---|---|
| PGS2 | 50 €, reviewed every year by the assembly. |
| PGS3 | 50 € |
| PGS4 | 50 € |
| PGS5 | Between 5 € and 30 €, depending on the member's economic potential. |
| PGS6 | 30 € |
| PGS7 | Individuals: 30 €<br>Associations and entities: 100 € |
| PGS8 | 1 € per week |

Source: Compiled by the authors.

In PGSs, the fees paid by the members are constant; the fee is the same regardless of activity, size or sector. The fee paid is more a reflection of the concept of belonging to the initiative. Some of the cases studied establish self-managed complementary sources of funds such as canteens or catering services in the public activities they organise. In third-party certification systems, the fee corresponds to a payment for a service provided. Accordingly, it is priced by considering different aspects: size, type of activity, variety of activities. There is a fixed minimum rate established regardless of the size of the farm or unit, although, in this case, there is a significant difference when the entities are public (lower price) or private (higher price). There is also a difference in the degree of transparency. The two public certification bodies publish all their fees on the webpage, while the private body has published no specific figures and only responds when asked for a price quote.

### 4. Discussion

Firstly, it is clear that in terms of the stages and activities carried out in building trust, PGSs and third-party guarantee systems are not that different. Both share the following basic stages: a visit to the farm; evaluation of the visit, with the participation of someone who was not involved in the visit; a final decision regarding authorisation of certification in the case of third-party or membership in the case of PGSs, after the trial period. In addition, both systems have documentation, such as introductory letters of commitment, guides for visits or reports on visits. So, both systems have the same initial vision from the point of view of conception and design of the procedure for building trust. Trust is built through in situ assessment of the farm, peer review of the results of this activity, and monitoring of the documents required. We begin to question whether PGSs are as profoundly ground-breaking relative to the official status quo as [29] and [4] suggest they are.

Nevertheless, in studying how each of these activities is carried out, the differences become more apparent, specifically in the nature of the tools implemented, the governance structure and the commitment required from producers and consumers.

Third-party certification systems include more elements in the procedure, generally associated with more paperwork. In contrast, PGSs require much less paperwork associated with the guarantee. The design and content of these documents, moreover, is decided by members of the initiatives (producers, consumers and, sometimes, directly connected technical staff), which is why the language and formats reflect the local environment. This contrasts with the technical and legal-administrative language in documents used by third-party entities.

Secondly, the third-party certification system includes highly rigorous checking procedures such as analytics, associated with increased costs. As noted by numerous authors [3,5,9,17], the distinct approaches of the two systems are reflected in the differences seen in the levels and meanings of the fees payable. In third-party certification, they are

simply a payment for services, and the amount depends on the characteristics of the farm/producer (size of the farm, variety and type of activities to be certified, etc.); in PGSs, you pay for membership of the group, to be part of it, and the amount is fixed.

In all the PGSs studied, the fees are lower than the minimum costs of third-party certification. It could be said, therefore, that PGSs are more accessible to all types of producers and do not penalise productive diversity (combining different productive sectors in the same farm) or pluriactivity (for example, combining agricultural activity and processing) with an increase in costs.

These differences may be attributed to the PGSs' simple, accessible procedures in terms of paperwork (field notebooks) and costs (analytics) [4,9,30], which are two aspects of third-party certification that are heavily criticised [3,6,17]. Nevertheless, despite the costs of analytics, a significant percentage of PGSs include them as part of their checking procedures. In several of the PGSs studied, analytics were included if covered by external support (for example, a university), but not if they had to be paid for by the stakeholders. Consequently, although analytics are used where possible, this tends to be carried out only when there is external funding.

Another essential differentiating element is the nature of the visits in each system. In third-party certification, technical audits have an exam format, checking if the producer complies with the established criteria and issuing a pass or fail. The certification bodies, by law, are prohibited from adopting an advisory role during this activity [2,24]. In PGSs, visits are meetings among peers. In addition to the checks on whether the activity qualifies, information and knowledge are exchanged between peers and other stakeholders present (consumers and/or technicians). This is reflected in the recommendations commonly made, also identified in other contexts such as Japan [24]. These visits are then evaluated both on their current standards and whether and how the recommendations made on previous visits were acted upon, and how the farm has been evolving. This means that PGS visits drive the agroecological transition on farms; regular visits by other farmers are a valuable tool in fostering the agroecological transition [31].

Where we find a yawning gap between PGSs and the official status quo, as [29] and [4] stated, is in their governance system and its collective nature. We argue that the fundamental difference between the two systems centres on the people responsible for each activity and procedure. This is directly related to the demands both systems require of the producers in terms of commitment and involvement.

In the third-party system, the organisation hires technical staff, often from outside the context and geographical area, to conduct assessments. The consolidation of the retail food industry and the higher private retail standards have also had profound effects. They have led to a reconfiguration of social, political and economic relations throughout the global agri-food system, which is now regulated mainly by the power of the supermarkets [32]. However, in PGSs, the members themselves assume this responsibility, sharing a social and geographical context and in turn issuing a guarantee through membership of the system (in the case of the producers). This is standard in all PGSs where systems have been established in different countries, such as Italy, Mexico, Peru, Japan, and Brazil [1,3,4,6,9,24,33]. Thus, producers (and consumers when they participate) take responsibility for building trust in the members of their own initiative. To do this, they must be fully involved, including making decisions related to every aspect of the process. This process contains procedures that develop capacities and abilities, building collective knowledge and empowering those social groups involved [2,11,33].

Since PGSs require collective spaces for discussion, construction and decision-making, they are mechanisms that build or strengthen links of collectivism in the local community [4,5].

An important discussion emerges from the previous differences: the requirements placed on the people in each guarantee system. Third-party systems do not call for any involvement or work from the producer in the guarantee process beyond receiving the visit of the technical inspector and keeping documentation up to date (field notebook).

The significant amounts of paperwork required can be outsourced to people with a technical background when this cost can be met economically. It is, therefore, a mechanism that is open and accessible to any individual producer who can pay the corresponding costs and take on or outsource the administration of the necessary paperwork.

In participatory guarantee systems, however, producers must become involved in the whole procedure and the associated tasks related to the guarantee and to the operation of the initiative, which is established as an association. Thus, all PGSs set up regular meetings (general assemblies or compulsory meetings) where decisions related to the operation of the PGSs are made, and commissions or working groups for specific tasks or internal processes. Participation in these activities cannot be outsourced but must be undertaken by members.

These requirements and operating mechanisms are seen in PGSs not only in Spain but also in other contexts studied (Brazil [34,35]; Italy [4], Peru [34,36]; Mexico [6,7]; Japan [24]. This leads us to conclude that the benefits for small and medium-sized productions of opting for PGSs as an alternative to third-party certification (lower costs and less administration) may be counterbalanced by the additional costs of the time and ongoing participation required by the PGS.

This could also explain some weaknesses identified in PGSs, which show significant differences between the theory and its practical application, and problems of reliability [1,6,7,9,10,34]. These difficulties may have arisen because the requirements demanded of members by PGSs—especially in activities and decision-making—were not clearly stated from the outset. Interestingly, in case studies where these weaknesses were identified, PGSs were driven by external organisations, i.e., they were not self-organised and managed by the producers themselves [1,6,33]. We, therefore, recommend that in contexts where external institutions aim to promote PGSs in local communities of producers (and consumers, where possible), an analysis should be conducted on the attitudes of participation by the people to be involved or at least to have procedures of self-organisation in place which guarantee that the mechanisms of the PGSs will be developed and adopted appropriately.

What is also questionable is the belief that PGSs are more accessible than third-party certification for small and medium-sized producers [4,9,11]. While there is no doubt that the costs and paperwork of the entry procedure set by third-party systems exclude these producers, the ways in which PGSs frame entry procedures can also limit accessibility. Some filtering of potential applicants to the PGS may be caused by factors such as the role played by the Sponsor, attitudes towards opening the farm to visits from members and the procedure of making initial contact through people who are already part of the initiative. Thus, the entry mechanisms of some PGSs mean that potential applicants who have not had previous contact with them or their members must seek them out. Moreover, some farmers could be deterred by the PGS's requirement of transparent and open attitudes vis-a-vis the farmer and their activities. This is not the case in third-party certification, where the procedures facilitate access to any interested producer (as long as the associated costs and volume of paperwork are affordable and deemed worthwhile).

Finally, the following criticisms have been made of third-party guarantee systems, and this is where PGSs procedures may present themselves as an alternative: inefficiency and the possibility of fraud [4]; the annual visit being carried out by a technical person from outside the social context [2,12]; a bureaucratic and expensive operation which does not favour small and medium-sized productions [4,6,9]; the marginalisation of local knowledge and abilities in favour of technical ability and knowledge [15,24,37–39]; and the system's poor transparency and, therefore, the limited capacity for social control [2,24,40,41].

The personal commitment required by PGSs of its members also reflects the vision of alternative agri-food networks, where aims are identified which go beyond shortening food chains and which place at their centre the value of local community and trust

[19,20,36,42–44]. These systems are designed for people with a collective and political view of agri-food systems, in line with the political dimension of Agroecology [45] as proposed by [3]. This would therefore be a prerequisite to making PGSs' procedures work (leaving aside certain structural weaknesses which may also affect their success, such as those noted by [1,14,33] or [11]). Without this collective support, PGSs would likely have weak structures and procedures which would necessitate external support for their correct functioning, running the risk of constructing the usual small certifiers, as pointed out by [1].

## 5. Conclusions

One of the main conclusions of this study is that systems are not significantly different in either the stages they follow or the frameworks of their procedures. Both focus on visits, reviews of visits, review of documentation and more than one filter being added at the final decision stage. In addition, while regulated procedures are more systematic and uniform in the various third-party certification bodies, it is interesting that PGSs share many procedures despite not being officially regulated. Furthermore, private certification systems are less transparent than public certification ones. This is an important consideration when the context calls for confidence building in a sector and something the EU and the member states should consider when privatising the organic certification system.

In addition, some procedures in PGSs are not so defined or consensual, which make PGSs adaptable to groups and specific social and geographical contexts and allow people at all levels in the PGSs to participate. However, this can also make it more difficult to obtain recognition, both from institutions and from communities and society in general. This needs to be considered carefully if and when official regulations contemplate recognising PGSs as a valid confidence-building system for organics.

The procedures adopted by PGSs require people to take on responsibilities and tasks that in third-party systems are delegated to an intermediary not involved in the context. It takes a particular type of person who is willing to become involved in these kinds of procedures, and not every social group can develop these collective procedures.

The issue that particularly concerns us is the way in which the European organic regulations seem to discourage and even penalise self-management processes, even though they are in the minority and create high-quality socially innovative processes (which are not openly acknowledged). The lack of support from European public policies towards sustainable agri-food systems is reflected in the lack of policy integration, as identified by many academics (Cf. [46]).

However, we also acknowledge the risks of proposing official recognition for, and the instrumentalisation of, systems that require a willingness to actively participate and convictions of a collective nature and seek to build responses through complex local social processes.

PGSs represent profoundly democratic food governance systems and are not, therefore, systems applicable to every context or social group; however, much external support is provided, as they would likely lose the collective identity and self-management which they have so far enjoyed, at least in Spain.

The results of this study suggest that PGSs are not a response to third-party certification but a social construct with their identity rooted in their internal diversity and which respond to a political vision different from that of third-party systems. While PGSs engage in dialogue with third-party systems, they involve social innovations of such magnitude that they could only really be developed with a grassroots political approach. With this approach, the mechanisms developed, and the underlying rationale behind them would be closely linked. In addition, the lack of wider awareness of PGSs and their position on the fringe of official status leads to unverified statements from stakeholders in the organic sector suggesting that, because they are non-regulated (their heterogeneity of proposals, each one doing what they want to), their trustworthiness is

lower. Thus, the PGSs' starting point is the demand for recognition of their fundamental links with the political dimension of agroecology.

For this reason, it may also be inappropriate to use an 'instruction manual approach' to establishing PGSs in contexts open to the idea out of necessity—impoverished producers, small farmers, difficulty accessing the market—since the dedication required in terms of both time and convictions may not be feasible.

Our analysis leads us to conclude that the governance model of PGSs, unlike third-party certification systems, favours collaboration among stakeholders for the construction and development of the system itself and of more solid Alternative Food Networks (AFNs). In this model, all the following factors become relevant: the decentralisation of decision-making, an orientation towards community links, an organisational structure that emphasises participation and commitment, the development of shared interests and responsibilities, the ability to interact with a variety of central stakeholders in the food system (e.g., consumers, technical staff, activist groups and social movements), and communication between them in terms of coordination and trust.

Lastly, taking into account the costs of both systems and the service-provision logic of third-party certification, it must be said that the latter favours neither the productive diversity inherent in the agroecological vision nor moves towards economic cooperation. Therefore, the logic which recognises economic interdependence must be emphasised. It is reflected in the conceptualisation of PGSs' fees, based on mutual care and resilience and on the inclusion of productive projects and the local community, rather than on penalising diversity.

Taking into account the evidence presented on the differences and procedural particularities of guarantee systems by third parties and the PGSs, we consider that PGSs can solve some exclusion problems created by third-party certification. Moreover, they generate locally adapted individual and collective processes in each territory. Thus, given their collective and participatory nature, they are not homogenisable, and they are applicable only in certain contexts or productive projects.

**Supplementary Materials:** The following supporting information can be downloaded at: www.mdpi.com/article/10.3390/su14063325/s1, Table S1. Documentation consulted in each case study and communication with key informants; Table S2. Development of tasks in the entry procedure of both guarantee systems, and the agents responsible; Table S3. Development of tasks in the follow-up procedure of both guarantee systems, and the agents responsible

**Author Contributions:** Conceptualization, methodology, and formal analysis M.C.-P. and I.H.-P.; investigation and writing—review and editing, M.C.-P., I.H.-P. and M.B.-Z.; visualisation, project administration, funding acquisition, M.C.-P. and I.H.-P.; supervision M.C.-P. All authors have read and agreed to the published version of the manuscript.

**Funding:** This research was funded by ERDF/Ministry of Science, Innovation and Universities – Spanish Research Agency / Project reference: CSO2017-85660-R.

**Institutional Review Board Statement:** Not applicable.

**Informed Consent Statement:** Informed consent was obtained from all subjects involved in the study.

**Data Availability Statement:** Not applicable.

**Conflicts of Interest:** The authors declare no conflict of interest. The funder had no role in the design of the study; in the collection, analyses, or interpretation of data; in the writing of the manuscript, or in the decision to publish the results.

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
