# Peer review of "Participatory Guarantee Systems: When People Want to Take Part"

_sustainability, doi:10.3390/su14063325_

Round 1

Reviewer 1 Report

The manuscript addresses a qualitative study of differences and similarities between how two types of certification systems work. The work in question is relevant, and I see it as an important contribution to the field. Thus, in my opinion, it deserves publication.

The introduction is concise and adequate for the theme. The objective is clear.The results section is long, with several tables that can be optimized or made available as supplementary material.

As a significant limitation, even dealing with a qualitative study, we have a small number of third-party certifications and limited access to their information. 

Author Response

Thanks very much for your time and dedication to help us improve our article. Thanks you also for your kind words, that we have very much appreciated. We have made the following changes, trying to answser to your concerns.

Comment 1.- The results section is long, with several tables that can be optimized or made available as supplementary material

Answer: Thanks for this contribution. We have moved original tables 3, 6 and 7 to supplementary materials, in order to shorten the results section. As they were supported by the figures, we think not important information misses with this change.

Comment 2.- As a significant limitation, even dealing with a qualitative study, we have a small number of third-party certifications and limited access to their information.

Answer: We agree with you, but we think there are two arguments that can defend the research as it is. First, third party certification procedures are officialy regulated in the European Union by the norm UNE-EN-ISO/IEC 17065. That is, even if there are many certification entities operating in the territory, they all are regulated and accredited following this norm. They do not have the possibility to do it in another way. The three certification entities results obtained in the research show that there are no differences between them, supporting the above argument. We have added a few sentence to clarify this (line 164).

Best Regards

Reviewer 2 Report

Esteemed Authors,

It has been a great honor, as well as a pleasantly challenging activity, to review the article entitled ”Participatory Guarantee Systems: When people want to take part.”

Agricultural and food production has significantly changed in the last 20 years. Obviously and logically, increasing population and consumption put pressure on the world's food supply. Accordingly to the data of the Food and Agriculture Organisation (FAO), by the effect of demographic growth and changes in diets and incomes, the demand for food will likely grow by 70% until 2050. The current outlook of the increasingly global market is marked by considerable uncertainties of supply linked to unpredictable economic, political, climatic, and biological developments. This implies a need for accelerated agricultural production growth in developing countries.

The list of new challenges is open and includes the most unexpected situations, from new crop and animal diseases to significant climate change and emerging diseases.

Current agriculture consumes enormous resources for development: over 70% of freshwater reserves are used for agriculture. At the same time, agriculture represents an essential threat to the environment: it is responsible for polluting almost 80% of the oceans and freshwater reserves. On the other hand, products of animal origin are among the top products in terms of greenhouse gas emissions, and approximately one-third of the food produced in the world for human consumption — nearly 1.3 billion tons — gets lost or wasted every year.

With a few exceptions, food production is growing. However, the challenges to primary production are increasing. Concerning organic farming in the European Union, the demand for such products is constantly growing, and the implementation mechanisms need to be continuously adjusted.

The recent reforms of CAP (Common Agricultural Policy) and other EU policies and international and bilateral trade negotiations take into account the objective of global food security. The  Joint Research Center (JRC) of the European Commission is involved in the impact assessment of policies regarding food security. Also, the potential trade agreements through economic modeling and the global CGE (Computable General Equilibrium) models assess the economy-wide impacts of the trade policy changes. All these changes are affecting all sectors of the partners. Besides, the global partial equilibrium models simulate the consequences incurred by the agricultural areas of the partners.

The situation is much more complicated concerning organic products than conventional products. Given that organic products are among the top products subject to various frauds, consumer confidence appears as an essential element of the development of the food chain.

Theory-wise, the paper is likely to elicit specialists' interest in consumer behavior, organic food production, organic food consumption, sociology, sustainable development of agriculture, agricultural productivity, public policies, and public health. The paper presents essential practical applicability primarily related to the sustainable development of agriculture, consumer education, organic food, public policies, and social awareness of farmers' management in crop production.

The paper is well structured and possesses an appreciable novelty character. The main components of the article – Introduction; Context and Methods; Results; Discussion and Conclusions - are organized judiciously and directly linked to one another.

The documentation is adequate, and the provided scientific results are precise. The goal of the conducted research is well specified and delineated. The working protocol is appropriate, and the used analysis methods are coherent with the proposed objectives.

The bibliography of the paper is generous. What is even more relevant for the overall quality of the article, despite the appreciable number of bibliographic sources, all the authors in the bibliographic reference list are quoted in the text of the material (without exception).

The article is very well documented, and most bibliographic references are recent and very recent.

I would advise the authors to be more careful concerning the bibliography: it is preferred to mention the authors in alphabetical order, and references without specified authors are cited at the end of the list of references in chronological order. I also recommend using a single system not only in citations but also when it comes to journals. I am referring here mainly to mentioning the following elements for each article consulted: journal, volume, issue, and pages. Supplementary, the DOI may also be noted if the authors desire, but the essential descriptive elements are the previously mentioned ones.

Also, to avoid confusion, it is recommended to accurately mention the article's descriptive elements - for example - the additional mention of the article number, where the situation requires it.

For example - page 20, lines 650-651, number 38 in the bibliographic references list – Bain C., Ransom E., Worosz M.R. Constructing Credibility: Using Technoscience to Legitimate Strategies in Agrifood Governance. Journal of Rural Social Sciences (or ISO Abbreviation – J. Rural Soc. Sci.) 2010, 25, 3, Article 9, 1-33; DOI: https://egrove.olemiss.edu/jrss/vol25/iss3/9.

The work also benefits from adequate iconographic support, materialized by ten tables and three figures. The data included in the tables accurately reflect the main objectives and the results obtained.

The authors should pay more attention to the use of certain abbreviations to avoid confusion; basically, all abbreviations are to be used in the text-only after at least one mention made in extenso.

The obtained results are interpreted correctly, and their practical value is visible.

The graphical representation of the results is adequate; as for the paper's grammar, the text is very well written. Consequently, I have only a few recommendations, as follows:

Page 8, line 227 – replace “in their being” with “in them being”;

Page 17, line 502 – replace “adopted by PGSs requires” with “adopted by PGSs require”.

Minor corrections and clarifications notwithstanding, the authors’ work and obtained results are highly commendable. They add significant value to the paper and may constitute a launching pad for further valuable studies.

Provided that the authors verify the paper and perform the required corrections, the article can be accepted and published in the Sustainability.

Best Regards,

Reviewer

Author Response

Thanks a lot for your contributions around the paper’s issue, and thanks a lot for your kind words, that we have highlyt appreciate. We have introduce the following changes, trying to answer to your observations and suggestions.

Comment 1.- I would advise the authors to be more careful concerning the bibliography: it is preferred to mention the authors in alphabetical order, and references without specified authors are cited at the end of the list of references in chronological order. I also recommend using a single system not only in citations but also when it comes to journals. I am referring here mainly to mentioning the following elements for each article consulted: journal, volume, issue, and pages. Supplementary, the DOI may also be noted if the authors desire, but the essential descriptive elements are the previously mentioned ones. Also, to avoid confusion, it is recommended to accurately mention the article's descriptive elements - for example - the additional mention of the article number, where the situation requires it.

Answer.- Thanks for the suggestion. However, we have followed the journal instructions in ordering the references as they are, and also giving for each one of them te information that appears, so we guess we must leave them as they are. Nevertheless, your comment invited us to review the references list and we have identified two mistakes that have been corrected (reference 21 and reference 46).

Comment 2.- The authors should pay more attention to the use of certain abbreviations to avoid confusion; basically, all abbreviations are to be used in the text-only after at least one mention made in extenso.

Answer: thanks for the comment. We have reviewed the text and identified this mistake at the Author contributions section. We have there introduced the extended version of the abbreviations that were not previously used.

Comment 3.- Page 8, line 227 – replace “in their being” with “in them being”;

Answer: We have changed it. Thanks!

Comment 4.- Page 17, line 502 – replace “adopted by PGSs requires” with “adopted by PGSs require”.

Answer: We have changed it. Thanks!

Best Regards